**Data Availability Statement:** All relevant data are within the manuscript and its Supporting Information files.

# Patterns of mobility and its impact on retention in care among people living with HIV in the Manhiça District, Mozambique

**Edson L. Bernardo**[1,2], **Tacilta Nhampossa**[2,3], **Kate Clouse**[1,4], **James G. Carlucci**[1,5], **Sheila Fernández-Luis**[2,6], **Laura Fuente-Soro**[2,6], **Ariel Nhacolo**[2], **Mohsin Sidat**[7], **Denise Naniche**[2,6], **Troy D. Moon**[1,5]*

1 Vanderbilt Institute for Global Health, Vanderbilt University Medical Center, Nashville, Tennessee, United States of America, 2 Manhiça Health Research Center, Manhiça, Mozambique, 3 National Institute of Health of Mozambique, Maputo, Mozambique, 4 Vanderbilt University School of Nursing, Nashville, Tennessee, United States of America, 5 Division of Pediatric Infectious Diseases, Department of Pediatrics, Vanderbilt University Medical Center, Nashville, Tennessee, United States of America, 6 Barcelona Institute for Global Health, ISGlobal, Hospital Clinic, Universitat de Barcelona; Barcelona, Spain, 7 Faculty of Medicine, University Eduardo Mondlane, Maputo, Mozambique

* troy.moon@vumc.org

## Abstract

### Introduction

Retention in HIV care is a challenge in Mozambique. Mozambique´s southern provinces have the highest mobility levels of the country. Mobility may result in poorer response to HIV care and treatment initiatives.

### Methods

We conducted a cross-sectional survey to explore the impact of mobility on retention for HIV-positive adults on ART presenting to the clinic in December 2017 and January 2018. Survey data were linked to participant clinical records from the HIV care and treatment program. This study took place in Manhiça District, southern Mozambique. We enrolled self-identified migrants (moved outside of Manhiça District ≤12 months prior to survey) and non-migrants, matched by age and sex.

### Results

390 HIV-positive adults were included. We found frequent movement: 45% of migrants reported leaving the district 3–5 times over the past 12 months, usually for extended stays. South Africa was the most common destination (71%). Overall, 30% of participants had at least one delay (15–60 days) in ART pick-up and 11% were delayed >60 days, though no significant difference was seen between mobile and non-mobile cohorts. Few migrants accessed care while traveling.

### Conclusion

Our population of mobile and non-mobile participants showed frequent lapses in ART pick-up. Mobility could be for extended time periods and HIV care frequently did not continue at

**Funding:** This research was supported by the Fogarty International Center of the National Institutes of Health under Award Number D43 TW009745, awarded to TM. The content is solely the responsibility of the authors and does not necessarily represent the official views of the National Institutes of Health. The funders had no role in the study design, data collection and analysis, decision to publish, or preparation of the manuscript.

**Competing interests:** The authors have declared that no competing interests exist.

the destination. Studies are needed to evaluate the impact of Mozambique´s approach of providing 3-months ART among mobile populations and barriers to care while traveling, as is better education on how and where to access care when traveling.

## Introduction

Retention in HIV care and treatment continues to be a significant challenge in Mozambique since the inception of its HIV programs in the early 2000s [1–6]. Mozambique is one of the countries most affected by HIV with a prevalence of 13.2% in 2015 and 1.2 million persons receiving antiretroviral therapy (ART) as of December 2017 [1–7]. Significant efforts have been made to address the HIV epidemic in Mozambique, yet as of March 2019 only 49% of persons living with HIV (PLHIV) were receiving ART and only 77% of those on treatment had undetectable viral loads [8].

Retention in care and treatment is essential to maintaining good adherence to ART and ensuring optimal treatment outcomes [9]. Various studies have identified barriers to HIV retention in Mozambique, including medication side effects, complexity of dosing schedules, low patient education level, poverty, stigma, distance and transportation challenges, quality of care received, and reliance on traditional medicines [2,5,10,11]. Mozambique´s continued low retention threatens to undo the gains made in terms of HIV testing and enrollment over the last several years [12].

One potential challenge to retention in HIV care and treatment that is gaining attention is the mobility of people between regions and countries [13–17]. In 2011, UNAIDS and the International Organization for Migration (IOM), signed an agreement recognizing HIV-related challenges faced by mobile populations [18]. Their objective was to integrate human rights and the needs of mobile populations into national and regional HIV responses and to ensure universal access to HIV prevention, treatment, care and support [18].

Recent studies have shown that the mobility of PLHIV can create potential vulnerabilities for the spread of new HIV infections and may result in poorer response to HIV care and treatment initiatives, leading to an overall negative impact on the health and economic status of this population [16,19,20]. One study in Lesotho, looking at barriers to access of PLHIV migrating to South Africa, found that roughly 25% had defaulted on their treatment, with the primary reason reported as a failure to get back to Lesotho to collect their medications due to fears that treatment received in South Africa would be "stronger" and "have more side effects" [20].

The southern provinces of Mozambique have some of the highest mobility levels of the country. This region borders both South Africa and Eswatini and features two high-traffic transport corridors which link Maputo with these two countries. Studies conducted in 2012 showed that these transport corridors had high levels of commercial sex work and that high-risk sexual practices were common along these routes [21]. To date, there have been no studies specifically describing the implications of mobility on HIV care in Mozambique.

We sought to describe the patterns of mobility among PLHIV seeking care at the Manhiça District Hospital (MDH) in southern Mozambique and the potential impact on engagement in HIV care. Results from this analysis may be of importance to Mozambique and other countries in the region with large migratory populations as they strive to improve effectiveness of HIV care and treatment service approaches to this population.

## Materials and methods

### Setting

This study was conducted at the Manhiça Health Research Center (*Centro de Investigação de Saúde de Manhiça*, CISM) at MDH of Manhiça District, Mozambique [22]. At the time of the study, the district population was served by MDH, one rural hospital and 18 peripheral health posts. HIV services included HIV care and treatment clinics, HIV counseling and testing, as well as prevention of mother-to-child transmission (PMTCT) services. As of 2016, Mozambique had implemented "test and treat" strategies for same-day initiation of ART once someone is identified as being HIV-positive [23].

Manhiça District is a rural district located in southern Mozambique, whose catchment area has a high prevalence of both tuberculosis (TB) and HIV infection. Agriculture and subsistence farming predominate, but large portions of the population engage in seasonal migration to South Africa, seeking work in the mines. A strength of Manhiça District is that more than half of the population (approximately 210,000 inhabitants) participates in a continuous health and demographic surveillance system (HDSS), carried out by CISM [24]. The HDSS collects data on all vital events such as births, deaths, pregnancies and mobility patterns for those registered in the system [25–27].

### Study design and participants

We conducted a cross-sectional survey to collect demographic information and mobility history, including whether migration was international or domestic, rural or urban, the duration of travel, pattern of home returns, whether children accompanied the travel, healthcare access while at their destination, and self-reported adherence to ART. Consecutive patients, ≥18 years old, presenting for their scheduled HIV clinic visit at MDH from December 01, 2017 to January 31, 2018 were invited to participate. These months were chosen due to the fact they are holiday months in which many migrating patients return to Manhiça. Patients were first approached for inclusion in this study if they were residents of the District of Manhiça, had migrated out of the district in the 12 months prior to study enrollment, and had initiated ART prior to study enrollment. For each person enrolled with a history of migration, an age (± 5 years)- and sex-matched person without a history of migration, and who attended the HIV clinic was invited to participate. Controls were selected through convenience sampling from a list of patients seen at the clinic within the seven-day period following recruitment of the patient with a history of migration. As this study recruited participants from a single district in which migratory patterns historically predominated to South Africa, our results may not be representative of migratory HIV-infected adults residing elsewhere in Mozambique or the region.

### Data collection and management

Patients attending routine scheduled visits for HIV care were screened for eligibility by a project-dedicated health assistant after completion of their routine clinic visit. Surveys were administered face-to-face in the preferred language of the participant, either Portuguese or a local language (Changana). Survey responses were translated in real-time by the interviewer and recorded electronically in Portuguese via a tablet utilizing Open Data Kit software 1.4 (ODK) [28]. Data were then uploaded into Research Electronic Data Capture (REDCap), a secure web-based application designed to support data capture for research studies [29]. Data were then linked with the patient´s clinical information registered within the HIV programs

electronic patient tracking system (ePTS), including sociodemographic variables, clinical history, and pharmacy/ART pick-up data.

## Definitions utilized

Individuals were identified as *mobile* if they answered "yes" to the survey question "Have you moved outside of Manhiça District in the 12 months prior to survey implementation?". Per national protocols at the time of the study, HIV patients seen at MDH were typically dispensed a 30-day supply of their ART and other HIV medications. While clinic visit schedules could be spaced out on an every 1-month, 3-month, or 6-month basis depending on the needs of the individual, generally the patient, or a designated representative, had to return to the clinic once a month for pharmacy refills.

For our outcome of interest, patients were defined as "*retained in care*", if they had no missed clinic visits or pharmacy pick-ups of >15 days during the 12-months prior to survey implementation. If the patient had at least one delay of between 15–60 days in picking up their ART, they were defined as "*delayed ART pick-up*". If the patient had a least one delay of >60 days in their ART pick-up they were defined as "*lost-to-follow-up*", regardless of the fact that they all were back in care at the time of completing the survey. These definitions are the official definitions used by the Mozambican Ministry of Health to assess patient retention in HIV care and treatment programs.

## Statistical analysis

Participant recruitment was conducted in the months of December and January and based on prior clinic visit volumes it was anticipated that MDH would see approximately 100 daily patient visits during this time frame in the HIV care and treatment program and that 20% of these visits would meet eligibility criteria as a participant with history of migration out of the district (~15 patients per day or a total of 150 mobile participants). As our estimated recruitment sample was fixed (based on convenience), the statistical power to detect a difference in LTFU was variable depending on actual LTFU rates in each group (i.e. for a LTFU of 20% in the non-mobile group, we would expect a 96% power to detect a difference if the LTFU was 40% in the migrant group, but only a 46% power to detect a difference if the LTFU was 30% in the migrant group).

Data were analyzed using Stata version 15.1. Descriptive statistics were used to summarize the data, using frequency distribution tables. Associations between categorical variables were tested using Chi-square or Fisher's exact tests. A p-value of less than 0.05 was considered statistically significant.

## Ethical considerations

This study received ethical approval from the Institutional Committee for Bioethics in Health of CISM (CIBS-CISM/169/2017) and from Vanderbilt University Institutional Review Board (#200328). Written informed consent was provided by all participants and approvals were obtained from the Provincial Directorate of Health.

## Results

A total of 390 mobile and non-mobile HIV-positive adults on ART were included in this study. The majority (72%) were 25–44 years old and 57% were male. Two-thirds (66%) of participants were married. Only 19% had formal education beyond primary school. The majority (87%) were employed, with a significantly higher proportion of employment being reported in

our mobile population than the non-mobile population (98% vs. 78% respectively, p<0.001). Overall, 95% of participants reported ownership of a cellphone. Seventy-seven percent reported being on ART for 1–5 years. Most (66%) were WHO clinical stage I at the time of survey administration. Finally, 76% of our study population had disclosed their HIV status to their partner or family (Table 1).

Among mobile study participants, 68% reported leaving the district 2–5 times over the 12-month period prior to survey administration, with 38% reporting they were away 15 days to 3 months each trip, and another 25% reporting they were away 3–9 months each trip (Table 2). Nearly 80% reported their migration destination as being an urban setting. For those who were mobile within Mozambique, roughly 83% stayed in the southern region, migrating to either the capital city of Maputo (45%) or its surrounding areas within Maputo Province (15%) or Gaza Province (23%), all within a 1-2-hour drive of Maputo. For those reporting migrating to a location outside of Mozambique, 97% reported South Africa as the country of destination. Seventy-six percent of participants reported work as the reason for travel, with another 14% reporting they were seeking better life opportunities.

Overall, only 68% of our mobile population reported that they had access to ART when traveling. When questioned as to the reason for no access to ART, 26% said they did not look for places within their destination location to receive ART, while 68% responded "other" without further explanation as to the reason. Further, 62% of participants reported that a family member brought them their ART when they were traveling and another 37% reported they took enough ART with them to last the length of their trip. Additionally, 98% of our mobile population reported they did not seek HIV care at their site of dislocation. We then asked how frequently our mobile population was able to maintain clinic visits (regardless of what was scheduled) at MDH during their travel periods. The most common response was every three months (34%), followed by monthly (30%) and then every six months (26%).

In order to understand if there were any differences for migrants within Mozambique versus those who migrate outside the country, we asked questions on self-reported perceived challenges in accessing HIV care (Table 3). No statistically significant difference was seen between the two groups in relation to their responses, however important challenges for our cohort as a whole were identified. When we asked about challenges in accessing HIV care upon their return to Manhiça District, the majority (~90%) reported no challenges. However, roughly 8% reported that "long wait times at the clinic" was a barrier to accessing care upon their return. We then asked about alternative ART dosing schedules that could help facilitate their adherence to their medications. Most (64%), stated they would prefer a >3-month dosing schedule, followed by a 3-month dosing schedule (33%). Twenty-seven percent of participants who migrate out of the country stated they had a relationship partner at their destination location, compared to only 19% for those migrating within Mozambique, though this was not statistically significant. Finally, approximately 15% of our mobile population reported they had children at their destination location, regardless if within Mozambique or not.

We explored the participant´s perceptions as to whether they had ever been classified as lost to follow-up since they initiated taking of ART. Overall, 5% of the mobile population responded that they had ever been lost to follow-up, compared to only 1% of our non-mobile population (p = 0.036) (3% of the total population). Next, we analyzed clinical data based on pharmacy pick-up dates for both our mobile and non-mobile participants. Forty-one participants had insufficient pharmacy pick-up data to be included in this analysis. Of the 349 participants that had complete data, 30% had at least one delay (15–60 days late) in ART pick-up documented in the pharmacy records for the 12-month period prior to survey administration, and 11% had at least one documented delay in ART pick-up of >60 days in ART. There was

**Table 1. Sociodemographic characteristics of mobile and non-mobile HIV-positive adults enrolled in HIV care and treatment at Manhiça District Hospital.**

| N = 390 | Mobile n (%) | Non-mobile n (%) | Total n (%) | p-value* |
|---|---|---|---|---|
| **Total** | 195 (50) | 195 (50) | 390 (100) | – |
| **Age (n = 389)** | | | | |
| 18–24 | 4 (2) | 10 (5) | 14 (4) | 0.571 |
| 25–34 | 59 (30) | 54 (28) | 113 (29) | |
| 35–44 | 84 (43) | 86 (44) | 170 (43) | |
| 45–54 | 32 (18) | 33 (17) | 69 (18) | |
| ≥55 | 12 (6) | 11 (6) | 23 (6) | |
| **Gender** | | | | |
| Male | 116 (59) | 108 (55) | 224 (57) | 0.671 |
| Female | 79 (40) | 87 (45) | 166 (43) | |
| **Marital status** | | | | |
| Married | 129 (66) | 129 (66) | 258 (66) | 0.041 |
| Widowed | 14 (7) | 27 (14) | 41 (11) | |
| Divorced/Separated | 25 (13) | 25 (13) | 50 (13) | |
| Single/Never Married | 27 (14) | 14 (7) | 41 (11) | |
| **Education** | | | | |
| No education | 18 (9) | 32 (16) | 50 (13) | 0.012 |
| Grade 7 or less | 140 (72) | 126 (65) | 266 (68) | |
| Completed High school | 36 (19) | 29 (15) | 65 (17) | |
| Some College | 1 (<1) | 8 (4) | 9 (2) | |
| **Religion** | | | | |
| Catholic | 88 (45) | 93 (48) | 181 (46) | 0.764 |
| Protestant Christian | 40 (21) | 43 (22) | 83 (21) | |
| Islamic | 6 (3) | 5 (3) | 11 (3) | |
| Zion | 53 (27) | 47 (24) | 100 (26) | |
| Other | 8 (4) | 7 (4) | 15 (4) | |
| **Employment (n = 343)** | | | | |
| No | 3 (2) | 42 (22) | 45 (13) | <0.001 |
| Yes | 145 (98) | 153 (78) | 298 (87) | |
| **Type of employment (n = 298)** | | | | |
| Administrative | 6 (4) | 17 (11) | 23 (8) | <0.001 |
| Agriculture | 8 (6) | 15 (10) | 23 (8) | |
| Self-employed (non-farming) | 26 (18) | 39 (25) | 65 (22) | |
| Industry/Mining | 18 (12) | 10 (7) | 28 (9) | |
| Domestic work | 12 (8) | 27 (18) | 39 (13) | |
| Construction | 23 (16) | 14 (9) | 37 (12) | |
| Other | 53 (36) | 30 (20) | 83 (28) | |
| **Owns cellphone (n = 389)** | | | | |
| Yes | 191 (98) | 177 (91) | 368 (95) | 0.003 |
| No | 4 (2) | 17 (9) | 21 (5) | |
| **Time since HIV diagnosis** | | | | |
| Less than 3 months | 2 (1) | 1 (1) | 3 (1) | 0.450 |
| 3 to 12 months | 12 (6) | 7 (4) | 19 (5) | |
| 1 to 5 years | 149 (76) | 146 (75) | 295 (76) | |
| More than 5 years | 32 (16) | 41 (21) | 73 (19) | |
| **Duration on ART (n = 389)** | | | | 0.425 |
| Less than 3 months | 2 (1) | 1 (1) | 3 (1) | |

*(Continued)*

**Table 1.** (Continued)

| N = 390 | Mobile n (%) | Non-mobile n (%) | Total n (%) | p-value* |
|---|---|---|---|---|
| 3 to 12 months | 14 (7) | 8 (4) | 22 (6) | |
| 1 to 5 years | 149 (77) | 149 (76) | 298 (77) | |
| More than 5 years | 29 (15) | 37 (19) | 66 (17) | |
| **WHO stage at diagnosis (n = 357)** | | | | |
| I | 117 (66) | 117 (65) | 234 (66) | 0.1384 |
| II | 26 (15) | 29 (16) | 55 (15) | |
| III | 28 (16) | 28 (16) | 56 (16) | |
| IV | 6 (3) | 6 (3) | 12 (3) | |
| **HIV status disclosed to partner** | | | | |
| Yes | 152 (78) | 145 (74) | 297 (76) | <0.001 |
| No | 17 (9) | 36 (18) | 53 (14) | |
| Not applicable (no partner) | 26 (13) | 0 (0) | 26 (7) | |
| Refused to answer | 0 (0) | 14 (7) | 14 (4) | |

*Chi-square or Fisher's exact tests.

ART = antiretroviral therapy.

no significant difference in delays noted between our mobile and non-mobile cohorts (Table 4).

## Discussion

Manhiça District, in southern Mozambique, is a region with high HIV prevalence (~40%) and a large population that migrates to South Africa every year to work in the mines [30]. Furthermore, Mozambique, more broadly, is a country that continues to have challenges with the long-term retention of PLHIV in treatment [1–6]. As such, we sought to better understand the dynamics of this mobile HIV population and to explore the impact that their mobility has on retention and adherence to HIV care and treatment.

Through purposefully enrolling a mobile population, we found that they were predominantly male and largely of the 25-44-year-old age group, with traveling for work-related reasons being the most commonly cited reason for travel outside of Manhiça District. This younger, male demographic makes up a large proportion of the labor workforce in Africa and is consistent with prior descriptions of cross-border migration for employment within the Southern Africa Development Community (SADC), that has been going on for more than 150 years [31–33].

Among the mobile participants in our study, multiple migration events each year were common and often for considerable amounts of time. Two-thirds of the migrant cohort reported that they had traveled outside of the district between 2–5 times over the course of the 12-month period prior to survey implementation and one-quarter (25%) staying away 3–9 months per trip. South Africa was the most common destination location for our cohort of patients, however a quarter of our mobile HIV population had migratory destinations inside of Mozambique, mostly to urban areas within a 1-2-hour drive of Maputo. In recent years, Mozambique's economy has experienced dynamic growth, mainly as a result of the expansion of extraction industries for coal, natural gas, petroleum, and gemstones [34]. Future studies should be established to better document and understand this in-country migration, including ways to reduce its potential impacts on retention in HIV care and adherence to ART medication.

**Table 2. Migration patterns and characteristics of mobile HIV-infected adults enrolled in HIV care and treatment at Manhiça District Hospital.**

| N = 195 | n (%) |
|---|---|
| **Number of times left district (over the last 12 months)** | |
| Once | 18 (9) |
| 2 times | 48 (25) |
| 3–5 times | 84 (43) |
| Once a month | 18 (9) |
| Once a week | 6 (3) |
| Other | 5 (3) |
| Don't know | 16 (8) |
| **Typical length of time at destination of travel** | |
| Less than a week | 27 (14) |
| Less than 15 days | 25 (13) |
| From 15 days to 3 months | 75 (38) |
| From 3 to 9 months | 49 (25) |
| More than 9 months | 15 (8) |
| Don't Know | 4 (2) |
| **Where traveled to?** | |
| Rural | 37 (19) |
| Urban | 155 (79) |
| Both | 3 (2) |
| **Which province (if within Mozambique) (n = 63)** | |
| Maputo City | 24 (38) |
| Maputo Province | 8 (12) |
| Gaza | 12 (19) |
| Inhambane | 1 (2) |
| Sofala | 1 (2) |
| Nampula | 3 (5) |
| Tete | 3 (5) |
| Don't know | 11 (17) |
| **Which country (if outside Mozambique) (n = 132)** | |
| South Africa | 128 (97) |
| Botswana | 3 (2) |
| Democratic Republic of Congo | 1 (1) |
| **Reason for traveling (n = 193)** | |
| Work | 147 (76) |
| Death in Family | 2 (1) |
| Looking for better life opportunities | 27 (14) |
| Following Partner | 10 (5) |
| Others | 7 (4) |
| **What kind of job at destination? (n = 148)** | |
| Administrative | 6 (4) |
| Agriculture | 8 (5) |
| Self Employed | 26 (18) |
| Industry/Mining | 18 (12) |
| Domestic Work | 12 (8) |
| Unemployed | 3 (2) |
| Other | 75 (51) |

*(Continued)*

**Table 2.** (Continued)

| N = 195 | n (%) |
|---|---|
| **Access to ART at destination** | |
| Yes | 133 (68) |
| No | 62 (31) |
| **Reason for no access to ART at destination (n = 62)** | |
| Didn't look for it | 16 (26) |
| Didn't know could pick up at destination | 1 (2) |
| Denied follow-up at clinic | 1 (2) |
| No ART stock | 2 (3) |
| Other | 42 (68) |
| **How accessed ART when at destination (n = 133)** | |
| Sent by family member | 82 (62) |
| Local pharmacy | 1 (<1) |
| Took enough with them from Manhiça | 49 (37) |
| Other | 1 (1) |
| **Frequency of HIV care follow-up at destination** | |
| Monthly | 3 (1) |
| Every 2 months | 1 (<1) |
| Every 6 months | 1 (<1) |
| No follow-up | 190 (97) |
| **Frequency of HIV care follow-up (at MDH) after traveling?** | |
| Monthly | 59 (30) |
| Every 2 months | 16 (8) |
| Every 3 months | 66 (34) |
| Every 6 months | 51 (26) |
| Once a year | 1 (<1) |
| No follow-up | 2 (1) |

ART = antiretroviral therapy.

We found no significant difference between our mobile and non-mobile patients in terms of their likelihood to have been classified as delayed in their ART pharmacy pick-up or as lost-to-follow-up. This is likely due to the fact that actual LTFU rates in each group were lower than what was anticipated pre-study implementation and as a result we may not have been sufficiently powered to detect a difference. Lapses or delays in care were frequent, regardless of migrant status. Among all study participants, pharmacy records showed that 30% had at least one instance over the course of the preceding 12-month period in which they were delayed between 15–60 days, and 11% had been classified as being lost-to-follow-up (delay of >60 days in picking up their ART) at least once, a striking disparity to the 3% of participants who self-perceived they had ever been lost to follow-up. It is unclear whether the lower perceived lost to follow-up by self-report is based on response bias in which the participant responds in the way they feel the interviewer wants them to, or if this is a true lack of awareness as to what good adherence and retention really means. Further exploration of this phenomena is warranted. Furthermore, recruitment into this study was done during routine HIV clinic visits at MDH. Therefore, all participants in this study were somehow engaged in care or had re-entered care if they had previously been classified as being late to a visit or lost. As such, the proportion of our patients classified as "delayed" and "lost-to-follow-up" likely is an underestimation of the true retention for this region, as those who did not return to care are not captured here.

**Table 3. Self-reported perceptions of mobile HIV-positive adults about their HIV care by those who migrate within Mozambique versus outside of Mozambique.**

| N = 185 | In-country n (%) | Other Country n (%) | Total n (%) | P-value |
|---|---|---|---|---|
| **Total** | 53 (29) | 132 (71) | 185 (100) | – |
| **Difficulties in ART care when returning to MDH (n = 185)** | | | | |
| Long wait time at clinic | 4 (8) | 11 (8) | 15 (8) | 1.000 |
| Bad service | 0 (0) | 2 (1.5) | 2 (1) | |
| No difficulty found | 49 (92) | 117 (89) | 166 (90) | |
| Other | 0 (0) | 2 (1.5) | 2 (1) | |
| **Preferred dosing schedule when traveling (n = 183)** | | | | |
| 3-month supply | 16 (31) | 45 (34) | 61 (33) | 0.855 |
| 3 to 6-month supply | 25 (48) | 58 (44) | 83 (45) | |
| 6-month supply | 9 (17) | 25 (19) | 34 (19) | |
| Fine with current 1-month supply | 2 (4) | 3 (2) | 5 (3) | |
| **Do you have a relationship at destination location? (n = 185)** | | | | |
| Yes | 10 (19) | 35 (27) | 45 (24) | 0.273 |
| No | 43 (81) | 97 (73) | 140 (76) | |
| **Do you have children at destination location (n = 185)** | | | | |
| Yes | 7 (13) | 20 (15) | 27 (15) | 0.821 |
| No | 46 (87) | 112 (85) | 158 (85) | |

*Chi-square of Fisher's exact tests.

• MDH = Manhiça District Hospital.

• We had incomplete response data from 10 mobile participants that were excluded from this analysis.

Despite the lack of difference in overall retention of our mobile versus non-mobile HIV patients, there are some notable findings of our mobile population related to their HIV care and treatment that necessitate further exploration. First, even though our mobile population reported frequent migratory events over the course of a year and that each event could last from 15 days to 9 months, there was a nearly universal reported lack of HIV follow-up care at their site of destination. When asked about access to their ART medications at their destination, most respondents reported that they brought enough medications with them or that they had family members bring their ART to them while they were away. Additional research to verify and further explore these findings and other health-seeking behaviors of migrant PLHIV are warranted.

The lack of HIV treatment follow-up while traveling in our cohort, is consistent with a report looking at the challenges faced by migrant populations throughout South Africa, which

**Table 4. Comparison of antiretroviral therapy (ART) spacing pick-ups between mobile and non-mobile HIV-infected adults enrolled in HIV care and treatment at Manhiça District Hospital.**

| N = 349[&] | Mobile n (%) | Non-mobile n (%) | Total n (%) | p-value* |
|---|---|---|---|---|
| **Delayed ART pickups (15–60 days late)** | | | | 0.114 |
| Yes | 50 (29) | 55 (31) | 105 (30) | |
| No | 121 (71) | 123 (69) | 244 (70) | |
| **Lost to follow-up (>60 days)** | | | | 0.091 |
| Yes | 19 (11) | 18 (10) | 37 (11) | |
| No | 152 (89) | 160 (90) | 312 (89) | |

[&]41 participants did not have sufficiently complete pharmacy pick-up data to have been included in the analysis.

*Chi-square or Fisher's exact tests.

found that a lack of familiarity with a "foreign" health system; a perception that the South African health system would refuse care to non-citizens; a perception that South African ART is stronger and could have more adverse side effects; and a general preference to seek care when back at their home location, were all reasons listed for not seeking HIV care and treatment while in South Africa [16,20,35]. Our survey was not detailed enough to ascertain whether Mozambique´s migratory population would have similar responses. However, the overwhelming lack of follow-up care at the destination reported in our population highlights an urgent need to better understand this dynamic. Initiatives are needed to help migrants in overcoming the barriers to care and treatment when in a different country and provide education and preparatory guidance prior to one´s travel to ensure they either have what they need, or have a plan for how to get what they need when traveling. Our population´s self-reported perception of preparedness contradicts the pharmacy pick-up records that showed that >40% of our cohort experienced at least some form of delay in receiving their ART over the course of the 12-month period.

Twenty-four percent of our mobile population reported their migration destination to be within Mozambique. The challenges to accessing care described above for South Africa should not be present in another location within Mozambique, yet we found no real differences in responses for those migrating domestically compared to internationally with regards to accessing HIV follow-up care at the destination; accessing ART medications at the destination; or in the proportion of persons with pharmacy pick-up records documenting at least one period of delay in pick-up. This further highlights the need for future research into the barriers of accessing care and for retention in HIV care and treatment for Mozambique´s mobile population.

At the time of survey implementation, Mozambique´s national HIV treatment guidelines called for monthly pharmacy pick-up of ART, despite recent initiatives for spacing-out clinical visits to 3–6 months [36]. Later in 2018, after the current period of study, Mozambique began to allow 3-month dispensing of ART. Similarly, Mozambique could explore allowing patients who are stable, to receive a 6-month supply of their ART medications. The impact of dispensing longer periods of ART on retention in care, has yet to be fully examined but should be included in future studies.

A strength of this study is that we were able to triangulate participant survey data with their HIV pharmacy pick-up records in roughly 90% of participants, thus allowing us to make direct comparisons between actual pharmacy visits and one´s self-reported perceptions of their adherence. However, this also represents a study limitation as 41 participants had incomplete data related to their pharmacy pick-ups. This study also had several other limitations. First, patients were recruited from those attending an HIV clinic visit at MDH. This means that all patients enrolled in this study had either never left care or had returned to care if previously identified as delayed or lost-to-follow-up, thus likely giving an underestimation of the true poor retention of this population. Next, study interviews were conducted only in the months of December and January, based on an assumption that Manhiça's mobile HIV-infected population would return home during the holidays. This may not be accurate, resulting in our findings not being representative to this population as a whole. For this analysis our clinical data capture was limited to duration of ART treatment, length of time since HIV diagnosis, as well as WHO clinical staging information. We did not have access to other laboratory values that could be important in evaluating the effect of migration on retention in care, such as CD4 count and whether the participant was virally suppressed or not. Finally, we were unable to determine the exact amount of ART given to each patient at each of their pharmacy pick-up visits. At the time of survey implementation, Mozambican protocols limited dispensing of more than a 30-day supply of ART, however in practice there were likely instances where more than a 30-day supply was given, which could make some of our study participants appear

delayed or lost-to-follow-up based on visit schedules, when actually they had ART medication on them.

## Conclusion

Manhiça's HIV-positive mobile population frequented locations both within Mozambique and neighboring South Africa, characterized by multiple trips for work and being away for extended periods of time. A large proportion of both our mobile and non-mobile population experienced delays in ART pick-up during the prior 12-month period, despite self-perception of not being delayed. Challenges in accessing HIV care and treatment and decisions around accessing care when traveling were not different if migration was within Mozambique or out. Surprisingly, no difference in LTFU was found between our mobile and non-mobile populations, most likely due to being insufficiently powered to detect a difference based on lower than anticipated actual LTFU rates in each group. Further in-depth research with a larger population into the barriers of accessing care and for retention in HIV care and treatment for Mozambique´s mobile population is needed. Studies evaluating Mozambique´s relatively new initiative at providing 3-month supplies of ART are needed to evaluate its impact on retention and adherence. Alternative ART dosing strategies for this population may need to be for even longer. Finally, better education for mobile populations on how and where to access care when traveling is warranted.

## Supporting information

**S1 File. Study survey mobile adult english.**
(DOCX)

**S2 File. Study survey mobile adult portuguese.**
(DOCX)

**S3 File. Study survey nonmobile adult english.**
(DOCX)

**S4 File. Study survey nonmobile adult portuguese.**
(DOCX)

**S5 File. Deidentified data set.**
(XLSX)

**S6 File. Data dictionary for deidentified data set.**
(XLSX)

## Acknowledgments

The authors thank all study participants, the healthcare workers from CISM and MDH who assisted with data collection, and the district health authorities for their collaboration.

## Author Contributions

**Conceptualization:** Edson L. Bernardo, Tacilta Nhampossa, Sheila Fernández-Luis, Laura Fuente-Soro, Denise Naniche, Troy D. Moon.

**Data curation:** Tacilta Nhampossa, Sheila Fernández-Luis, Ariel Nhacolo.

**Formal analysis:** Edson L. Bernardo, Kate Clouse, James G. Carlucci, Laura Fuente-Soro, Mohsin Sidat, Denise Naniche, Troy D. Moon.

**Funding acquisition:** Troy D. Moon.

**Investigation:** Edson L. Bernardo.

**Methodology:** Edson L. Bernardo, Tacilta Nhampossa, Kate Clouse, James G. Carlucci, Sheila Fernández-Luis, Laura Fuente-Soro, Denise Naniche, Troy D. Moon.

**Project administration:** Denise Naniche.

**Supervision:** Tacilta Nhampossa, Mohsin Sidat, Denise Naniche.

**Validation:** Ariel Nhacolo.

**Writing – original draft:** Edson L. Bernardo.

**Writing – review & editing:** Edson L. Bernardo, Tacilta Nhampossa, Kate Clouse, James G. Carlucci, Sheila Fernández-Luis, Laura Fuente-Soro, Ariel Nhacolo, Mohsin Sidat, Denise Naniche, Troy D. Moon.

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
