## [Decision Letter · Decision Letter 0]

6 Nov 2020

PONE-D-20-25254

Patterns of mobility and its impact on retention in care among people living with HIV in the Manhiça District, Mozambique

PLOS ONE

Dear Dr. Moon,

Thank you for submitting your manuscript to PLOS ONE. After careful consideration, we feel that it has merit but does not fully meet PLOS ONE’s publication criteria as it currently stands. Therefore, we invite you to submit a revised version of the manuscript that addresses the points raised during the review process.

We look forward to receiving your revised manuscript.

Kind regards,

Matt A Price

Academic Editor

PLOS ONE

Journal Requirements:

2. Please provide additional details regarding participant consent. In the ethics statement in the Methods and online submission information, please ensure that you have specified (i) whether consent was informed and (ii) what type you obtained (for instance, written or verbal, and if verbal, how it was documented and witnessed). If your study included minors, state whether you obtained consent from parents or guardians. If the need for consent was waived by the ethics committee, please include this information.

3. In your Methods section, please provide additional information about the participant recruitment method and the demographic details of your participants. Please ensure you have provided sufficient details to replicate the analyses such as: a) a description of any inclusion/exclusion criteria that were applied to participant recruitment, and b) a statement as to whether your sample can be considered representative of a larger population."

4. Please include additional information regarding the survey or questionnaire used in the study and ensure that you have provided sufficient details that others could replicate the analyses. For instance, if you developed a questionnaire as part of this study and it is not under a copyright more restrictive than CC-BY, please include a copy, in both the original language and English, as Supporting Information.

5.During our internal checks, the in-house editorial staff noted that you conducted research or obtained samples in another country. Please check the relevant national regulations and laws applying to foreign researchers and state whether you obtained the required permits and approvals. Please address this in your ethics statement in both the manuscript and submission information. In addition, please ensure that you have suitably acknowledged the contributions of any local collaborators involved in this work in your authorship list and/or Acknowledgements. Authorship criteria is based on the International Committee of Medical Journal Editors (ICMJE) Uniform Requirements for Manuscripts Submitted to Biomedical Journals - for further information please see here: https://journals.plos.org/plosone/s/authorship.

6. Please provide a sample size and power calculation in the Methods, or discuss the reasons for not performing one before study initiation."

7. Please note that PLOS does not permit references to “data not shown.” Authors should provide the relevant data within the manuscript, the Supporting Information files, or in a public repository. If the data are not a core part of the research study being presented, we ask that authors remove any references to these data.

We ask that you please remove citations for unavailable and unpublished work, including manuscripts that have been submitted but not yet accepted (e.g., “unpublished work,” “data not shown”). Instead, include those data as supplementary material or deposit the data in a publicly available database.

8. Thank you for stating the following in the Acknowledgments Section of your manuscript:

'CISM is supported by the Government of Mozambique and the Spanish Agency for International Development

(AECID). ISGlobal acknowledges support from the Spanish Ministry of Science and Innovation

through the “Centro de Excelencia Severo Ochoa 2019-2023” Program (CEX2018-00806-S), and

support from the Generalitat de Catalunya through the CERCA Program.'

'The funders had no role in the study design, data collection and analysis, decision to publish, or preparation of the manuscript'

9. We note that you have indicated that data from this study are available upon request. PLOS only allows data to be available upon request if there are legal or ethical restrictions on sharing data publicly. For information on unacceptable data access restrictions, please see http://journals.plos.org/plosone/s/data-availability#loc-unacceptable-data-access-restrictions.

Additional Editor Comments (if provided):

[From the editor]

Methods, line 102: Please describe your statistical power to detect associations between your independent variables and your outcome of interest (retained in care, right?). With 195 persons in each comparison arm, are you well powered to see differences in retention in care, should they exist?

Methods, line 132: This is an outcome (singular) not “outcomes” (plural), right? Or am I misunderstanding something here?

Methods, line 143: “response” variables is what? The outcome (i.e., “retained in care” – one outcome variable?) Or did you also look at mobility? Also you may have gone between self report and pharmacy records; kindly make this more explicit.

Methods, line 143: do not report results in methods (i.e., move “we found no clinically or statistically significant findings” to the results section – is this what you show on line 187?)

Table 2: What is the “other” category that makes up more than two thirds of those didn’t/couldn’t get ART at travel destination? Is it for the reasons explained in the text? Consider adding this as a footer, or making more table rows to clarify.

Line 187 (and stat analysis in methods): is your multivariable analysis have “retention in care” as the outcome, or “self-reported perceived challenges in accessing HIV care”? Or both? Please clarify this in the methods, and do show these results, as no difference (assuming you have power to detect a difference should one exist) is an important result for all these analyses you may be doing.

Results, line 193: What’s the difference between “Most (45%), stated they would prefer a 3 to 6-month dosing schedule, followed by a 3-month dosing schedule (33%),”? These two responses seem to overlap/ include each other, making interpretation of this response challenging.

Line 201+ and Table 4: I am confused about this outcome. Why is the N for Table 4 only 349? I recommend considering a trichotomous outcome regarding retention in care: 1) retained in care, 2) Delayed ART pickups (15-60 days late) and 3) Lost to follow-up (>60 days) so that the entire cohort can be analyzed together. You are further impacting your statistical power by slicing up your study this way.

Table 4: If there is missing data (i.e., 349 vs. 390) please explain and show this so the reader can understand what’s missing and where.

Line 204: I am confused by your statement “This represents just 3% of our total population self-reporting they had ever been lost to follow-up (data not shown).” In table 4, it looks like you are showing the data: 37 report ever being lost to follow up, or 11% of the reduced cohort shown. This section is hard to follow, please revisit.

Discussion: please make note of how well powered your study was to detect a difference in your outcome (outcomes?), particularly in light of the fact that it appears your analysis was affected by missing data (e.g., Table 4 is not n=390)

Line 245: “However, what is striking is the large disparity between what we could document based on pharmacy pick-up records and the participants self-reported perception of their ever being lost to follow-up.” Did I miss this? Where in your results do you present this disparity? This seems like an important piece of data to show. This seems like an important data point to add, perhaps to Table 4. How did this outcome vary between self-report and pharmacy records, and how did this vary across study group (mobile, not-mobile).

Line 271: Kindly remind the reader that the aforementioned study (studies? Several references are listed) are from Lesotho as the next sentence otherwise is a bit confusing.

Reviewers' comments:

Reviewer's Responses to Questions

**Comments to the Author**

1. Is the manuscript technically sound, and do the data support the conclusions?

Reviewer #1: Partly

2. Has the statistical analysis been performed appropriately and rigorously? 

Reviewer #1: No

3. Have the authors made all data underlying the findings in their manuscript fully available?

Reviewer #1: Yes

4. Is the manuscript presented in an intelligible fashion and written in standard English?

Reviewer #1: Yes

5. Review Comments to the Author

Reviewer #1: Feedback to the authors

Major points

Abstract

Some important methodological details are not mentioned: self-reported interview data from mobile and non-mobile patients on ART care were compared with ART uptake data from clinical records, but the former comprised data from 390 participants whilst the latter was only available for 349 of these, thus possibly inviting selection bias. This fact is neither mentioned in the Abstract, nor the limitation discussed.

Methods

1. The statement that ‘consecutive’ patients were enrolled cannot be quite correct, given the matching process used. Please clarify whether all patients reporting to be mobile were consecutively enrolled. Regarding the matched non-mobile participants, please clarify whether they were selected from all those who met the matching criteria and presented to the clinic during the next 7 days. In other words, was the research assistant free to chose a patient who he/she preferred out of all possible matches, thus possibly inviting further selection bias, or was there a systematic rule for the selection that was followed?

2. As mentioned for the Abstract, in the section on Data collection the information is missing that clinical records were only available from subset of 349 of the 390 participants! This needs to be explicitly stated here or latest in the Results section; and an explanation provided on why this difference occurred. This is a limitation that should also be addressed in the Discussion.

3. The definition used for ‘delayed ART pick-ups’ (defined as gaps of more than 15 days after scheduled refills, see line 135) is rather inaccurate and broad, given that viral resistance often develops after much shorter interruptions in adherence. It would be helpful if the authors had studied whether a more stringent definition may be associated with a difference between mobile and non-mobile subgroups with respect to ART uptake.

Results

1. Some key information is missing that is required to understand the context: what is the total number of ART receiving patients registered at the Manhiça District Hospital from where the study population was selected (and of these what is the proportion of patients that achieved viral control at the time of the study)?

2. What are the proportions of study participants in the two comparison groups of this study who were virally controlled at their last visit? Was there a significant difference between study groups in this respect? Data on viral control were available at the time in a data set from another province of Mozambique that I have recently seen, but if such data were not available from the Manhiça HIV care centre at the time, what were the proportions of study participants in the two groups that had a satisfactory CD4 count (say e.g. >200 cells/mm3), and did this differ between study groups? This information would be important to assess the effects of mobility on retention in ART care, and would substantially increase the value of this publication. If no such data were available at all, this would at the very least require mention in the Discussion section.

3. Unfortunately there are many missing data without that this is explicitly stated (and without that the possible implications for the validity of the results are explored in the Discussion section). Examples include the following: (1) In Table 1, I was puzzled to realise that data on age was only available from 328 of 390 participants (16% missing!). How was it possible then to ‘match on age’? - (2) 23% of data were missing for ‘employment’, yet employment features as a key variable in the consecutive analysis. – (3) The one and only clinical information that related to clinical severity was WHO stage, however again data were missing for 33/390 participants (8%). Note that data on clinical severity may well be associated with ART uptake. – (4) Importantly as mentioned, data on ART uptake were missing for 41/390 participants (11%) from the hospital-based data set (Table 4). Further examples of missing data can be found in tables 2 and 3. With so many data missing, one wonders how valid are the results presented? The authors should at least indicate missing data in footnotes to the various tables, mention the more important data gaps when reporting results in the text, and reflect on this issue in the Discussion section.

4. In some cases, reported data do not tally within tables, or between tables and the text. Some examples: Table 1: employment data are 298, but 299 under ‘type of employment’. Table 2: 147 people travelled for work, but 148 of these gave information on what work they did. Also Table 2: 53 / 195 people travelled within the country (27%), but 25% are mentioned in the relevant text.

Discussion

The text is generally well written but the limitations described above (e.g. due to methodological issues and missing data) should be addressed.

Conclusion

The Conclusion section is misleading. It focuses on mobile HIV patients and the challenges that they encounter with respect to HIV care when they travel. This ignores that for the patients registered at the Manhiça Hospital the results underlying these conclusions seem to equally apply to the non-mobile group of patients. In fact, the lack of significant differences between the mobile and non-mobile groups with regards to ART uptake is surprising and should be mentioned as an important result in the Conclusion. The real conclusion should refer to the rather worrying lack of adherence in ART uptake that affects both groups.

General observations

The paper could be substantially shortened by condensing the text. Repeat statements should be deleted (e.g. see lines 262 and 277).

Minor points

1. Please use past tense consistently across the manuscript. (Occasionally the text alternates between past and present tense).

2. Some phrases are long-winded and could be compressed to just entail the essential information.

Methods

3. The section on data collection and management is NOT about data analysis as suggested by its headline.

6. PLOS authors have the option to publish the peer review history of their article (what does this mean?). If published, this will include your full peer review and any attached files.

Reviewer #1: No

---

## [Author Response · Author response to Decision Letter 0]

30 Mar 2021

please see uploaded document with our response to reviewers

1) Thank you for updating your Data Availability Statement as follows:

"The data underlying the results presented in the study are available here and from the Manhiça Health Research Center by request to Tacilta Nhampossa at tacilta.nhampossa@manhica.net"

-We also note your following additional comments:

"We have provided a de-identified data set to accompany this manuscript as supporting information."

1.1.) Please confirm whether the minimal data set is located in your manuscript and/or Supporting Information files. 

1.2.) If the minimal data set is located in your manuscript and/or Supporting Information files, please confirm whether we may update your Data Availability Statement as follows:

“All relevant data are within the manuscript and its Supporting Information files.”

Response: I confirm the data set is located in the manuscript and the data availability statement can be updated as proposed: “All relevant data are within the manuscript and its Supporting Information files.”

2) We note that you have replaced your Figure 1 map.

Before we can proceed, please clarify where the authors obtained the replacement map for Figure 1.

Response: We accessed a free generic map of Mozambique at https://map.comersis.com/Mozambique-maps-MZ.html and then the authors edited it to meet the needs of the figure for this publication. 

3) Thank you for providing the following response in your cover letter: "I am confused about your recent comment about funding. In the most recent versions submitted all reference to funding had already been removed from the acknowledgement section.

Please update the funding statement to the following:

This research was supported by the Fogarty International Center of the National Institutes of Health under Award Number D43 TW009745. The content is solely the responsibility of the authors and does not necessarily represent the official views of the National Institutes of Health. 'The funders had no role in the study design, data collection and analysis, decision to publish, or preparation of the manuscript"

Response: we confirm that the above is the funding statement as we would like it to be associated with this manuscript

Thank you as well for removing the following information from your Acknowledgements section: "CISM is supported by the Government of Mozambique and the Spanish Agency for International Development (AECID). ISGlobal acknowledges support from the Spanish Ministry of Science and Innovation through the “Centro de Excelencia Severo Ochoa 2019-2023” Program (CEX2018-00806-S), and support from the Generalitat de Catalunya through the CERCA Program."

However, before we proceed with updating your Financial Disclosure statement as requested, please clarify the following:

3.1. Who or what is meant by "CISM" and "ISGlobal"?

3.2. Were any authors of the study supported by the following: the Government of Mozambique and the Spanish Agency for International Development (AECID); the Spanish Ministry of Science and Innovation through the “Centro de Excelencia Severo Ochoa 2019-2023” Program (CEX2018-00806-S); the Generalitat de Catalunya through the CERCA Program?

If so, please provide the names of each author specifically associated with each funding source, as well as any relevant grant numbers associated with "the Generalitat de Catalunya through the CERCA Program".

3.3. Please clarify why the following information was omitted from your updated Financial Disclosure statement: "CISM is supported by the Government of Mozambique and the Spanish Agency for International Development (AECID). ISGlobal acknowledges support from the Spanish Ministry of Science and Innovation through the “Centro de Excelencia Severo Ochoa 2019-2023” Program (CEX2018-00806-S), and support from the Generalitat de Catalunya through the CERCA Program."

This information will be helpful in updating your funding statements.

Response: we apologize but there was initially confusion amongst the authors as to what constituted financial support for the work presented here. We have removed the above-mentioned reference because, while yes, they are funding or affiliations associated with either the institutions or individuals involved in this work, they did not specifically contribute to the work performed here in this manuscript. 

CISM is an acronym meaning Centro de Investigação em Saúde de Manhiça or Manhiça Health Research Center and it is affiliated with both the Government of Mozambique and the Spanish Agency for International Development (AECID), however neither directly funded this research project.

ISGlobal is the common name for the Barcelona Institute for Global Health, of which authors Fernández-Luis, Fuente-Soro and Naniche are affiliated. While yes, ISGlobal receives funding from the Spanish Ministry of Science and Innovation through the programs listed above, those programs should not have been referenced as funding the research presented here in this manuscript.

We again apologize for the confusion created, if any, but those statements should be removed.

---

## [Editor Report · Decision Letter 1]

15 Apr 2021

Patterns of mobility and its impact on retention in care among people living with HIV in the Manhiça District, Mozambique

PONE-D-20-25254R1

Dear Dr. Moon,

We’re pleased to inform you that your manuscript has been judged scientifically suitable for publication and will be formally accepted for publication once it meets all outstanding technical requirements.

Kind regards,

Matt A Price

Academic Editor

PLOS ONE
---

## [Editor Report · Acceptance letter]

14 May 2021

PONE-D-20-25254R1 

Patterns of mobility and its impact on retention in care among people living with HIV in the Manhiça District, Mozambique 

Dear Dr. Moon:

I'm pleased to inform you that your manuscript has been deemed suitable for publication in PLOS ONE. Congratulations! Your manuscript is now with our production department. 

Kind regards, 

on behalf of

Dr. Matt A Price 

Academic Editor

PLOS ONE